# Genome-Wide Identification of SnRK1 Catalytic α Subunit and FLZ Proteins in *Glycyrrhiza inflata* Bat. Highlights Their Potential Roles in Licorice Growth and Abiotic Stress Responses

**DOI:** 10.3390/ijms24010121

**Published:** 2022-12-21

**Authors:** Chao Yang, Guangyu Shi, Yuping Li, Ming Luo, Hongxia Wang, Jihua Wang, Ling Yuan, Ying Wang, Yongqing Li

**Affiliations:** 1Guangdong Provincial Key Laboratory of Applied Botany & Key Laboratory of South China Agricultural Plant Molecular Analysis and Genetic Improvement, South China Botanical Garden, Chinese Academy of Sciences, Guangzhou 510650, China; 2University of Chinese Academy of Sciences, Beijing 100049, China; 3Shanghai Key Laboratory of Plant Functional Genomics and Resources, Shanghai Chenshan Botanical Garden, Shanghai 201602, China; 4Key Laboratory of Crops Genetic Improvement of Guangdong, Crops Research Institute, Guangdong Academy of Agricultural Sciences, Guangzhou 510640, China; 5Department of Plant and Soil Sciences, University of Kentucky, Lexington, KY 40506, USA; 6Guangdong Provincial Key Laboratory of Digital Botanical Garden, South China Botanical Garden, Chinese Academy of Sciences, Guangzhou 510650, China

**Keywords:** SnRK1, FLZ, *Glycyrrhiza inflata*, stress response, growth

## Abstract

Sucrose non-fermenting-1-related protein kinase-1 (SnRK1) and its scaffolding proteins, FCS-like zinc finger proteins (FLZs), are well conserved in land plants and involved in various processes of plant growth and stress responses. *Glycyrrhiza inflata* Bat. is a widely used licorice species with strong abiotic stress resistance, in which terpenoids and flavonoids are the major bioactive components. Here, we identified 2 SnRK1 catalytic α subunit encoding genes (*GiSnRK1*α*1* and *GiSnRK1α2*) and 21 FLZ genes in *G. inflata*. Polygenetic analysis showed that the 21 GiFLZs could be divided into three groups. A total of 10 representative GiFLZ proteins interact with GiSnRK1α1, and they display overlapped subcellular localization (mainly in the nucleus and the cytoplasm) when transiently expressed in *Nicotiana benthamiana* leaf cells. Coinciding with the existence of various phytohormone-responsive and stress-responsive cis-regulatory elements in the *GiSnRK1*α and *GiFLZ* gene promoters, *GiFLZs* are actively responsive to methyl jasmonic acid (MeJA) and abscisic acid (ABA) treatments, and several *GiFLZs* and *GiSnRK1*α*1* are regulated by drought and saline-alkaline stresses. Interestingly, *GiSnRK1*α and 20 of 21 *GiFLZs* (except for *GiFLZ2*) show higher expression in the roots than in the leaves. These data provide comprehensive information on the SnRK1 catalytic α subunit and the FLZ proteins in licorice for future functional characterization.

## 1. Introduction

Licorice (Glycyrrhiza) belongs to the genus Leguminosa Glycyrrhiza linn, and there are around 20 species/subspecies worldwide [1,2]. Licorice is a group of a commercially valuable plants, with a wide range of uses in the pharmaceutical, tobacco, cosmetics, and food industries [3]. Owing to its beneficial effects for reducing toxicity and increasing the efficacy of other herbal medicines when used together, licorice is one of the most prescribed herbs [4,5]. Around 400 total bioactive compounds have been isolated from licorice [6,7]. For an example, glycyrrhizic acid, a kind of triterpenoid mainly accumulated in the roots and rhizomes [8], shows good inhibitory effects on HIV [9], hepatitis [10], leukemia [11], gastric cancer [12], and SARS-CoV-2 [13,14]; it is also commonly used as a spice and food additive because of its desirable sweetness (50–200 times that of sucrose) [15]. In China, the Glycyrrhiza species of medicinal licorice, including *G. uralensis* Fisch, *G. inflata* Bat., and *G. glabra* L. are mainly distributed in the northwest arid, semi-arid, and desert regions, and *G. inflata* has better tolerance to drought and soil saline-alkaline stresses than the other two species, in addition to most commercial crops from the same family, such as wild soybean [16,17,18,19]. Thus, *G. inflata* is not only an important economic crop in the pharmaceutical and industrial market, but also a good research material to excavate useful genes for the genetic breeding of crops with stress resistance.

Sucrose non-fermenting-1-related protein kinase-1 (SnRK1) is a heterotrimeric kinase complex consisting of catalytic α and regulatory βγ subunits [20,21,22], which play a vital role in the global control of cellular energy homeostasis, as well as stress responses [23,24,25]. *Arabidopsis thaliana* encodes three isoforms of the catalytic α subunit, namely AtSnRK1.1/AtSnRK1α1, AtSnRK1.2/AtSnRK1α2, and AtSnRK1.3/AtSnRK1α3 [25,26,27], but only *AtSnRK1.1* and *AtSnRK1.2* appear to be expressed, and *AtSnRK1.1* is responsible for most of the SnRK1 activity [25]. The double mutant of *AtSnRK1.1* and *AtSnRK1.2* is lethal, while overexpression of *AtSnRK1.1* leads to late flowering and enhanced tolerance of plants to nitrogen or carbon starvation, implying their important roles during plant growth and stress responses [23,24].

Recently, FCS-like zinc finger proteins (FLZs) have been identified as the scaffolding proteins of SnRK1, which directly interact with the subunits of SnRK1 [28,29,30,31]. Studies showed that FLZs are plant-specific regulatory proteins containing a common DUF581 (domain of unknown function 581)/FLZ domain, which widely exists in land plants [32]. Studies in Arabidopsis, wheat (*Triticum aestivum*), rice (*Oryza sativa*), and maize (*Zea mays*) showed that FLZ genes are involved in the regulation of plant growth and stress responses [28,33,34,35,36,37]. For instance, overexpression of *AtFLZ4/IRM*1 (*Increased Resistance to Myzus persicae 1*) increases the resistance to aphid attack, but leads to a significant reduction in plant growth in Arabidopsis [33]. Ectopic expression of the salt-inducible wheat *FLZ* genes *TaSRHP* and *TaFLZ1D* in Arabidopsis results in enhanced tolerance to salt stress [34,35]. Recently, Ma et al. systematically identified 29 FLZ genes in rice and proved that *OsFLZ18* negatively regulates rice growth under submerged conditions [36]. The maize genome contains 37 FLZ genes, and overexpression of ABA-inducible *ZmFLZ25* confers transgenic plants with higher sensitivity to ABA treatment [37]. However, the information regarding SnRK1 and FLZ genes has not yet been documented in medicinal plants.

To investigate the potential roles of SnRK1 and FLZ genes in medial plant, we performed a genome-wide survey in *G. inflata* and analyzed phylogenetic relationships, SnRK1α-FLZ interaction, subcellular localization, cis-regulatory elements, tissue-specific expression pattern, and their responses to MeJA, ABA, drought, and saline-alkaline treatments. These results established a foundation for further studying of their biological functions.

## 2. Results

### 2.1. Identification of SnRK1 Catalytic α Subunit and FLZ Proteins in G. inflata

In this study, we used the protein sequence of AtSnRK1.1 (AT3G01090.1) as a query to perform BLASTP against our sequenced *G. inflata* genome and successfully identified two SnRK1 catalytic α subunit encoding genes, named *GiSnRK1α1* and *GiSnRK1α2* (Table 1). Using the same strategy (the protein sequence of the FLZ domain in AtFLZ1 (AT5G47060.1) used as the query), 21 FLZ protein encoding genes were identified (Table 1), and named as *GiFLZ1* through *GiFLZ21*, based on their chromosomal localization (Appendix A). Detailed information about each *GiSnRK1*α and *GiFLZ* gene, including gene name, locus ID, chromosome, open reading frame (ORF) and protein length, predicted protein molecular weight (MW), isoelectric point (*p*I), and subcellular localization, is listed in Table 1. Their gene structures were shown in Appendix A. All *GiFLZ* genes contain 2 exons and 1 intron (Appendix A), the ORFs range from 348 to 1338 bp, and the encoded proteins contain 115 to 445 amino acids, with MW from 13.00 to 48.91 kDa, and the *p*I from 5.92 to 10.48 (Table 1). The WoLFPSORT prediction showed that GiSnRK1α proteins might be localized in the cytoplasm and the nucleus, while GiFLZ proteins mainly target the nucleus, cytoplasm, chloroplast, and mitochondria (Table 1).

### 2.2. Phylogenetic Analysis of FLZ Proteins from G. inflata and Arabidopsis thaliana

To investigate the phylogenetic relationships of *GiFLZ* gene family members, a total of 39 FLZ proteins from *G. inflata* (21) *and Arabidopsis thaliana* (18) were used to construct a neighbor-jointing phylogenetic tree. These FLZ proteins were roughly clustered into three major groups (groups I to III) (Figure 1A). Among them, group III contains the largest number of GiFLZs—up to 10—followed by Group I—containing 8—and Group II—containing 3—GiFLZs (Figure 1A). Multiple sequence alignment analysis confirmed that all of the 39 FLZ proteins have an FLZ domain sharing highly conserved sequence identity (CX_2_CX_3_LX_3-4_DX_3_YX_4-5_FCSX_2_CR) (Figure 1B).

### 2.3. Protein Interaction between GiSnRK1α1 and GiFLZs

Previous studies demonstrated that FLZ proteins could interact with SnRK1 catalytic α subunits in Arabidopsis, rice, and maize [28,29,30,31,36,37]. To determine whether GiFLZs also interact with GiSnRK1α, we selected ten GiFLZs from all three different groups: GiFLZ4 from Group I; (GiFLZ2, GiFLZ15/16—both CDS and protein sequences are identical with different chromosomal localization—Table 1 and Appendix A), GiFLZ18 and GiFLZ21 from Group II; and GiFLZ3, GiFLZ9, GiFLZ10, GiFLZ14, and GiFLZ17 from Group III (Figure 1A, marked with asterisks) to perform a yeast two-hybrid experiment with GiSnRK1α1 as the example (Figure 2). To this end, the full-length *GiFLZs* were fused to the GAL4 activation domain, while *GiSnRK1α1* was fused to the DNA binding domain, to produce AD-GiFLZ and BD-GiSnRK1α1 vectors, respectively. As expected, all the ten AD-GiFLZs interacted with BD-GiSnRK1α1, but not with the empty BD vector, as judged by the growth of the co-transfected yeast cells on selective medium plates (Figure 2). As shown in Figure 2A, GiFLZ2, 3, 4, 10, 14, 15/16, 18, and 21 showed strong interactions with GiSnRK1α1, where co-transformed yeast cell grew well on high strength selective medium plates (SD–Trp/–Leu/–His/–Ade, SD-4). However, GiFLZ9 and 17 showed weak interactions with GiSnRK1α1, and the yeast transformants could only grow on SD–Trp/–Leu/–His (SD-3) medium plates (Figure 2B).

### 2.4. Subcellular Localization of GiSnRK1α1-GFP and GiFLZs-GFP Fusions

Protein subcellular localization is important for understanding its function and aids the identification of potential interacting partners [38]. The WoLF PSORT prediction revealed that GiSnRK1α proteins might reside in the cytoplasm and the nucleus (Table 1). To validate this prediction, a fusion of *GiSnRK1*α*1* to *green florescence protein* (*GFP*) under the control of the *AtUBQ10* promoter was co-expressed with Histone3-mCherry (nuclear marker) in 6-week-old *N. benthamiana* leaf epidermal cells. The fluorescence signals indicated that GiSnRK1α1-GFP is localized to both the nucleus and the cytoplasm (Figure 3A), similar to that of AtSnRK1.1 [26,31].

Subcellular localization of GiFLZ proteins predicted by WoLF PSORT suggested that they were localized in diverse subcellular compartments, including the nucleus, cytoplasm, chloroplast, and mitochondria (Table 1). To confirm this result, the above selected 10 GiFLZs were fused with a GFP tag, and the resultant chimeric proteins were transiently co-expressed with Histone3-mCherry in *N. benthamiana* leaves. Confocal scanning microscopy revealed that the GFP signals of GiFLZ4, 9, 10, 14, 15/16, 18, and 21 fusions were found both in the nucleus and the cytoplasm, as was the case for GFP protein (Figure 3B). Interestingly, GiFLZ2-GFP and GiFLZ3-GFP were shown to be localized only in the nucleus (Figure 3C). The results are, in general, consistent with the bioinformatics prediction, in which they mainly localize in the nucleus (Table 1 and Figure 3). Furthermore, the overlapped distribution of GiSnRK1α1 and GiFLZs in the subcellular compartments allows for their potential interaction in plant cells.

### 2.5. Cis-Regulatory Elements in GiSnRK1α and GiFLZ Gene Promoters

Cis-elements in promoter regions are bound by different types of transcription factors to determine gene expression. We used the 2.0-kb genomic sequence upstream of the start code ATG of each gene to identify cis-elements against the PlantCARE database [39]. In total, 10 kinds well-studied stress-responsive and 12 kinds of phytohormone-responsive cis-elements were found (Figure 4 and Appendix A). As presented in Figure 4, each gene promoter contains 3–19 stress-responsive and 2–15 phytohormone-responsive cis-elements, respectively (Figure 4). In the stress-responsive group, anaerobic induction (ARE, GC-motif), drought-inducibility (MBS, DRE-core, DRE1), low temperature (LTR), stress (STRE, TC-rich repeats), and wounding responsiveness (WRE3, WUN-motif) elements were found (Figure 4 and Appendix A). In the phytohormone-responsive group, Me-JA (MYC, TGACG-motif), ABA (ABRE), salicylic acid (TCA-element), ethylene (ERE), auxin (TGA-element, AuxRR-core, TATC-box), and gibberellin (GARE-motif, P-box) elements were discovered (Figure 4 and Appendix A). These results suggest that the *GiSnRK1*α and *GiFLZ* genes might actively respond to various stresses and phytohormone signals, and that they can be regulated by different types of transcription factors.

### 2.6. Expression Pattern of GiSnRK1α and GiFLZ Genes in Response to MeJA and ABA Treatment

MeJA has been reported to regulate the biosynthesis of multiple specialized metabolites [40], for example, flavonoid and soyasaponin biosynthesis in *G. uralensis* [41,42], and 2 *GiSnRK1α* and 18 *GiFLZ* genes contain at least one JA-responsive cis-element (Figure 4). Therefore, we checked the responsiveness of *GiSnRK1*α and *GiFLZ* genes to MeJA treatment. As shown in Figure 5A, *GiSnRK1α2* showed obvious repressed expression, and 14 *GiFLZ* genes were MeJA-responsive, with 8 members being downregulated and 6 upregulated in hair roots with MeJA treatment for 2 h (Figure 5A). To further confirm this result, we performed a qRT-PCR assay. It is worth noting that, because of the high percentage of sequence identity of *GiFLZ13/14* (97.8%), *GiFLZ15/16* (100%), and *GiFLZ18/19* (98.2%) gene pairs, we could not distinguish them in the qRT-PCR assay, and the expression of *GiFLZ15/16* was too low to detect. As shown in Figure 5B, *GiFLZ2* was induced about 6-fold by MeJA application, whereas *GiFLZ3*, *4*, *9*, *10*, *17*, 18/*19,* and *21* were repressed, and *GiSnRK1α1* did not change significantly. Among them, the expression pattern of other genes in the qRT-PCR assay were consistent with the results obtained from the RNA-seq analysis, except for *GiFLZ3*, *4* and *13/14*. These results showed that most *GiFLZ* genes are MeJA responsive and might be involved in JA signaling to exert function in JA-regulated specialized metabolic processes.

ABA is the most important stress phytohormone [43], and 2 *GiSnRK1α* and 15 *GiFLZ* genes contain at least one ABRE element (Figure 4). To understand how the *GiSnRK1α* and *GiFLZ* genes respond to ABA, we performed qRT-PCR analysis in the roots of 5-day-old seedlings under 100 μM ABA treatment for 2 h. The results showed that ABA treatment could significantly induce the expression of *GiSnRK1α1*, *GiFLZ10*, *13/14*, and *21*, but repress those of *GiFLZ2* and *4*. The expression of *GiFLZ3*, *9*, *17*, and *18/19* showed no obvious change (Figure 5C).

### 2.7. Expression Patterns of GiSnRK1α and GiFLZ Genes in Response to Drought and Saline-Alkaline Stresses

Recent studies have suggested that FLZ and SnRK1 genes are involved in signaling and in the response to abiotic stimuli [36,37,44]. In the field, the most frequent abiotic stresses to licorice are drought and soil saline-alkaline stresses [16]. To uncover the potential roles of *GiSnRK1α* and *GiFLZ* genes in plant response to these stresses, we performed a qRT-PCR analysis in the roots of 5-day-old seedlings under drought (20% PEG6000) and mixed saline-alkaline (A5, B5 and C5, referred to Peng et al., 2008 [45]) treatments. After 2 h of 20% PEG6000 treatment, *GiFLZ2*, *3, 9,* and *10* showed a significant decrease in transcription, whereas *GiFLZ21* displayed a slight increase, and the others did not show any obvious change, when compared to the mock treatment (Figure 6A). With high salt treatment (A5), *GiFLZ2* was strongly induced by about six fold, while *GiFLZ3*, *9*, *13/14*, *18/19,* and *21* showed significantly decreased expression (Figure 6B), and the expression levels of *GiFLZ4*, *10, 17,* and *GiSnRK1α1* showed no change. Upon slight (B5) and moderate strength (C5) mixed saline-alkaline stresses, *GiFLZ2* was strongly induced; *GiFLZ10*, *17*, *19*, *21,* and *GiSnRK1α1* also showed increased expression (about two fold), while *GiFLZ9* and *13/14* obviously showed repressed expression (Figure 6C,D). These data indicated that the *GiSnRK1α1* and *GiFLZ* genes are actively responsive to the environmental stresses occurring in the natural habitat of *G. inflata*, and *GiFLZ2* showed the highest sensitivity to these stresses.

### 2.8. Expression Pattern of GiSnRK1α and GiFLZ Genes in Root and Leaf Tissues

Because the major beneficial compounds are accumulated more in roots than in the leaves, and the roots are the major medicinal elements [7,8], we checked the expression of the *GiSnRK1α* and *GiFLZ* genes in the roots and leaves. Interestingly, in our RNA-seq analysis, we found that these two *GiSnRK1*α, and most *GiFLZ* genes, exhibited greater expression in the roots than in the leaves, and this trend is more obvious in 2− and 3−year-old plants (Figure 7A). For instance, in 3-year-old plants, there are 19 *GiFLZ* genes showing obvious higher expression in the roots than in the leaves (Figure 7A). The only outlier is *GiFLZ2,* which displayed lower expression in the roots of 1- and 2-year-old plants, but comparable expression in roots and leaves in 3-year-old plants (Figure 7A). A similar expression pattern of FLZ genes in 2-year-old *G. uralensis* and *G. glabra* plants was also observed (Appendix A), indicating that this specific expression pattern of FLZ genes is conserved in different licorice species. To validate the results obtained from RNA-seq, we checked the gene expression of the selected *GiSnRK1α1* and *GiFLZs* with qRT-PCR assay, using roots and leaves harvested from 1-year-old *G. inflata* (Figure 7B). As shown in Figure 7B, only *GiFLZ2* showed lower expression in roots than leaves, in good agreement with the results of the RNA-seq assay, and all other *GiFLZ* genes and *GiSnRK1α1* showed higher expression in the roots than in the leaves. These results indicated that these *GiFLZ* and *GiSnRK1α* genes mainly function in the roots, and they might be involved in the biological processes occurring mainly in roots, i.e., glycyrrhizin biosynthesis and accumulation [8].

## 3. Discussion

SnRK1 is evolutionarily conserved in all eukaryotes, and FLZ genes first emerged in liverwort and were conserved in land plants [32]. Genome-wide identification and characterization of FLZ genes have recently been reported in Arabidopsis [31], rice [36], maize [37], and wheat [35]. However, they have not been studied in medicinal plants. Here, we report the identification of two SnRK1 catalytic α subunit encoding genes and 21 FLZ genes in *G. inflata* (Table 1). The Y2H assay showed that GiFLZs extensively interact with GiSnRK1α1 (Figure 2), similar to the cases observed in Arabidopsis, rice, and maize [28,29,30,31,36,37], indicating that the SnRK1α-FLZ module is also conserved in licorice. Subcellular localization analysis showed that GiSnRK1α1 and most examined GiFLZs are localized both in the nucleus and the cytoplasm, meeting the requirement for their interaction at the subcellular level (Figure 3). Furthermore, the gene expression patterns of GiSnRK1α1 and GiFLZs are similar and actively responsive to phytohormone or abiotic stress treatments (Figure 5, Figure 6 and Figure 7), meeting the requirement for their interaction at the tissue level. More work should be carried out to explore the biological functions of the GiSnRK1α-GiFLZ module in the future.

MeJA has been reported to regulate flavonoid and soyasaponin biosynthesis in *G. uralensis* [41,42]. In this work, we found that the gene expression level of *GiSnRK1α1* is not altered by exogenous MeJA treatment (Figure 5), but 14 of 21 *GiFLZ* genes are actively responsive to MeJA, and among them, 8 *GiFLZ* genes are downregulated, and 6 are upregulated (Figure 5), implying the critical roles of *GiFLZs* during licorice JA signaling or JA-mediated biological processes. In Arabidopsis, AtFLZs are found to interact with several transcription factors, including DELLA proteins [30], which interact with the core component in JA signaling MYC2 [46] to modulate anthocyanidin biosynthesis [47]. Therefore, in future, it will be interesting to explore the function of GiSnRK1α-GiFLZ modules in regulating plant specialized metabolism.

Wild *G. inflata* plants usually grow in arid, semi-arid, and saline-alkaline regions, and therefore, they have the ability to adapt to these harsh conditions [16,17,18,19]. Thus, *G. inflata* makes a good research material to explore the stress-resistant genes for crop breeding. In Arabidopsis, the ectopic expression of wheat *TaSRHP* and *TaFLZ1D* results in enhanced tolerance to salt stress [34,35], and the overexpression of *AtSnRK1.1* could confer resistance to drought stress [24]. Recently, a report proved that *LbSnRK1* positively regulates the tolerance of plants to high salt and drought stresses using the transgenic method in the sweet potato [48]. In this study, we found that the expression of *GiSnRK1α1* is induced by PEG and mixed saline-alkaline stresses (Figure 6), and most of the examined *GiFLZ* genes are also actively responsive to PEG and saline-alkaline treatments (Figure 6). ABA is called the “stress hormone,” and it is involved in the abiotic stress responses tested above. We checked the gene expression pattern under ABA treatment, and similar results were found (Figure 5C), indicating that *GiSnRK1α* and *GiFLZs* might also be involved in the stress responses in licorice. Among them, *GiFLZ2* is very special because it is strongly repressed by drought, but highly induced by salt and salt-alkali treatments (Figure 6), implying its important roles regarding the stress responses in licorice.

Interestingly, the RNA-seq data and qRT-PCR results showed that most *GiFLZ* genes and *GiSnRK1α* are highly expressed in the roots, and this trend is gradually enhanced following plant growth (Figure 7). Furthermore, similar expression profiles of FLZ genes were also observed in the other two Glycyrrhiza species, *G. uralensis* and *G. glabra* (Appendix A), indicating that the function of the FLZ genes may be similar in different licorice species. Since many active compounds accumulate specifically in licorice roots [7,8], the root-specific expression pattern of these genes points to their potentials in regulating specialized metabolism. In the future, it will be interesting to investigate the roles of *GiSnRK1α* and *GiFLZs* in regulating licorice growth and/or specialized metabolic processes.

## 4. Materials and Methods

### 4.1. Data Search and Analyses

The genomic data of *G. inflata* used in this study were sequenced and annotated by our lab (https://ngdc.cncb.ac.cn/;NGDC; CRA009044; accessed on 25 November 2022); *Arabidopsis* SnRK1α and FLZ data were downloaded from the TAIR database (https://www.arabidopsis.org/; accessed on 5 March 2022); The transcriptome data of roots and leaves or MeJA-treated hair roots have been deposited in the NCBI GEO repository (http://www.ncbi.nlm.nih.gov/geo; PRJNA574093; accessed on 25 September 2019). We used the BlastP program to search for all SnRK1α and FLZ proteins in the *G. inflata* genome with the BioEdit 7.2 software (https://bioedit.software.informer.com/; accessed on 10 March 2022). The protein sequences of AtSnRK1.1 (AT3G01090.1) and the FLZ domain of AtFLZ1 (AT5G47060.1) were used as queries to blast against the *G. inflata* genome (e ≤ 1 × 10^−5^, identities ≥ 0.5). The phylogenetic trees of the FLZ proteins from *A. thaliana* and *G. inflata* were constructed using the neighbor-joining (NJ) method by MEGA5.0 with a bootstrap 1000 [48]. The alignment of the protein sequences of the FLZ domain were conducted using MEGA5.0 software with the ClusterW program and edited with GeneDoc 2.7 (https://genedoc.software.informer.com/; accessed on 25 March 2022). The chromosomal locations of the *GiFLZ* genes were provided by the *G. inflata* genome data, and ShinyCircos software (https://shiny.hiplot.com.cn/shiny-circos/; accessed on 10 April 2022) was used to draw the location images. The WoLF PSORT web server (http://www.genscript.com/psort/wolf_psort.html; accessed on 10 May 2022) was used to predict the subcellular localizations of the GiSnRK1*α* and GiFLZ proteins, and the first two items are listed in Table 1. Gene structure information of each *GiSnRK1α* and *GiFLZ* was obtained from the generic feature format file of *G. inflata* genome data and TBtools (https://github.com/CJ-Chen/TBtools; accessed on 20 November 2022) was used to display their intron–exon structures. For cis-elements analysis, the promoter sequence (−2000 bp) of each gene was inferred on PlantCARE (http://bioinformatics.psb.ugent.be/webtools/plantcare/html/; accessed on 20 November 2022), and the stress-responsive and phytohormone-responsive cis-elements were recorded. The heat maps of gene expression profiles were generated with the OmicShare tools platform (http://www.omicshare.com/tools; accessed on 15 May 2022). 

### 4.2. Plant Materials and Stress Treatments

The roots and leaves of 1−, 2−, and 3−year-old licorice plants were collected in the test field at the Northwest Biological Agricultural Center (Ningxia, China) and flush frozen in liquid nitrogen. For the stress treatments, *G. inflata* seeds were surface sterilized and germinated on wet filter papers in darkness for 2 days at 25 °C, then transferred to 1/2MS plates and grown at 25 °C under a 16/8 h light/dark photo-cycle. After 5 days of growth, the seedlings were subjected to different treatments (water—mock, 100 µM MeJA, 100 µM ABA, 20% PEG6000, and saline-alkaline stresses, marked as A5, B5, and C5, all with salinity 160 mM and pH 7.26, 8.30, and 9.02, respectively, according to [45]). After 2 h, the roots of five seedlings were pooled as one sample, flush frozen with liquid nitrogen, and stored at −80 °C. All experiments were carried out with three independent biological replicates.

### 4.3. RNA Extraction and Quantitative Reverse-Transcription PCR (qRT-PCR) Analysis

Total RNA was extracted using the HiPure Total RNA Mini Kit (Code No. R4151-03, Magen, China). Then, 1 μg of total RNA was used for cDNA synthesis with the PrimeScript^®^ RT reagent Kit with gDNA Eraser (Cat No. RR047A, Takara, Dalian, China), according to the manufacturer’s instructions. The qRT-PCR reactions were performed in 384-well blocks using TB Green Premix ExTaq II (Tli RNaseH Plus) (Code No. RR820D, Takara, Dalian, China) on a LightCycler 480 (Roche, Switzerland), according to manufacturers’ instructions. The 2^−ΔΔCT^ method was used to normalize and calculate the expression levels of each tested gene relative to the internal reference gene (*GiCS*) [49,50]. All the primers used in this study are listed in Appendix A. Three biological replicates and three technical replicates were performed. Data were presented as the mean ± SD. Statistical analysis was performed in Microsoft Excel using TTEST at the 0.05 (*) and 0.01 (**) level of significance.

### 4.4. Subcellular Localization Assays in Nicotiana benthamiana Leaves

The subcellular localizations of GiSnRK1α and GiFLZ were tested by transient expression assays in 6-week-old *N. benthamiana* leaf epidermal cells. The full-length CDS of *GiSnRK1α1* and *GiFLZs* were cloned into the binary vector *pCambia1300-UBQ-GFP* in a frame with a *GFP* at the C-terminals, with the primers used listed in Appendix A. The fusion constructs were co-transfected into tobacco leaves with a nuclear localization marker (*Histone3-mCherry*) using an agroinoculation assay [36]. A confocal fluorescence microscope (Laica, SP8) was used to determine the localization of the target proteins 3 days after inoculation.

### 4.5. Yeast Two-Hybrid Assay

The yeast two-hybrid (Y2H) assay was performed following the manufacturer’s instructions (Clontech, http://www.clontech.com/; accessed on 12 March 2022) [36,37]. To generate the activation domain (AD)-fused GiFLZs, the full-length CDS of the respective genes were fused in-frame to the pGADT7 vector. For DNA-binding domain (BD)-fused GiSnRK1*α*1, the full-length CDS of *GiSnRK1α1* was PCR-amplified and separately cloned into the pGBKT7 vector. The primers used are listed in Appendix A. Yeast AH109 cells were co-transformed with each set of different vector combinations, as indicated. All yeast transformants were grown on the SD/−Trp−Leu (SD-2) medium and screened on the SD/−Trp−Leu−His (SD-3) or SD/−Trp−Leu−His−Ade (SD-4) medium for interaction tests.

## 5. Conclusions

In this study, we identified 2 SnRK1 catalytic α subunit encoding genes and 21 *GiFLZ* genes in *G. inflata*. The 21 GiFLZs could be roughly divided into 3 groups. A total of 10 representative GiFLZ proteins interacted with GiSnRK1α1, and they displayed overlapped subcellular localization. RNA-seq analysis showed that 20 *GiFLZ* genes were highly expressed in the root, and 14 members are responsive to MeJA treatment. Furthermore, qRT-PCR analysis showed that the expression levels of *GiSnRK1*α*1* and several *GiFLZ* genes (especially *GiFLZ2*) were significantly changed upon drought, saline-alkaline, or ABA stresses, highlighting their potential roles during the environmental adaptation of *G. inflata*. The comprehensive understanding of the *GiSnRK1α* and *GiFLZ* genes provides useful information for further functional studies to elucidate their regulation mechanisms and build a foundation for the future cultivation of high-quality cultivars of licorice through molecular breeding methods.

## Figures and Tables

**Figure 1 ijms-24-00121-f001:**
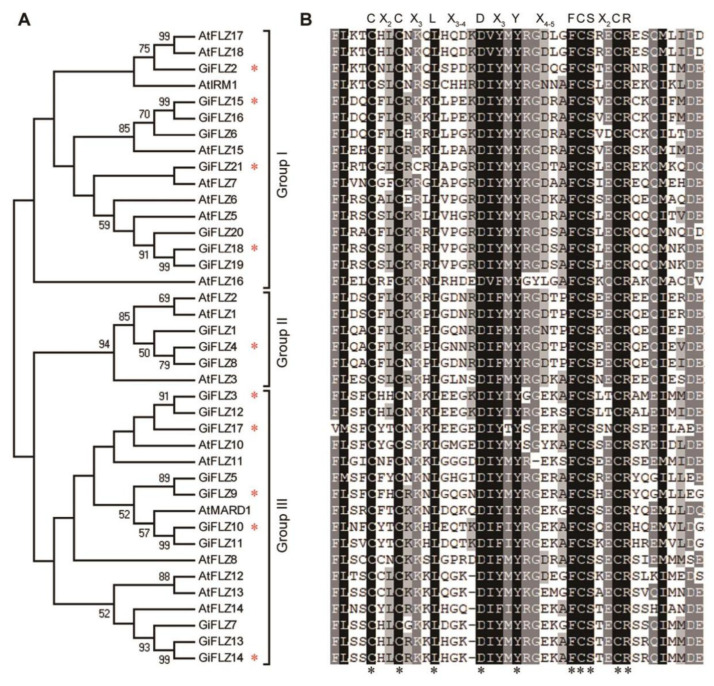
Phylogenetic tree and domain analysis of FLZ proteins. (**A**) Phylogenetic tree of FLZ proteins in *G. inflata* and *Arabidopsis thaliana*. The phylogenetic tree was constructed based on the multiple sequence alignments of FLZ proteins and generated with MEGA 5.0 software using the neighbor-joining method. The red asterisks marked the selected genes used in the subsequent study. (**B**) Protein sequence alignment of the FLZ domain. ClustalW was used for protein sequence alignment. The identical and similar amino acids in the FLZ domains were highlighted with dark and light gray backgrounds, respectively. The asterisks marked the conserved amino acids.

**Figure 2 ijms-24-00121-f002:**
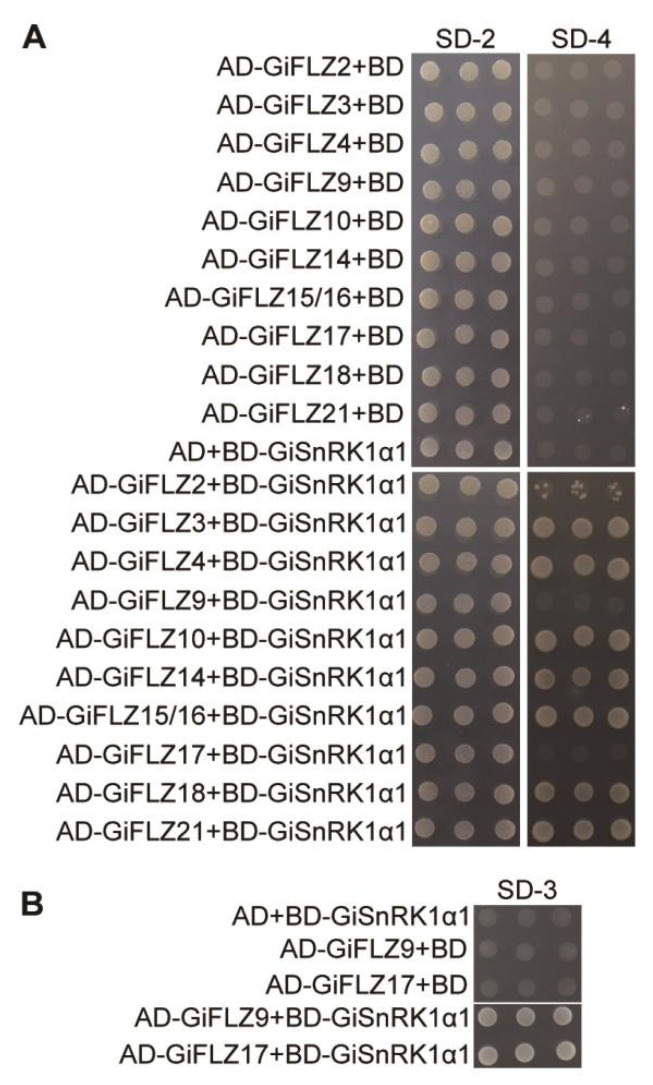
Ten selected GiFLZs interact with GiSnRK1α1 in yeast cells. (**A**,**B**) Interactions between GiSnRK1α1 and GiFLZs. GiFLZs were fused to the activation domain (AD), and GiSnRK1α1 was fused to the DNA-binding domain (BD), respectively. Co-transformed yeast clones were plated on SD/–Leu–Trp (SD-2), SD/–Leu–Trp–His-Ade (SD-4), or SD/–Leu–Trp–His (SD-3) plates to test the interaction.

**Figure 3 ijms-24-00121-f003:**
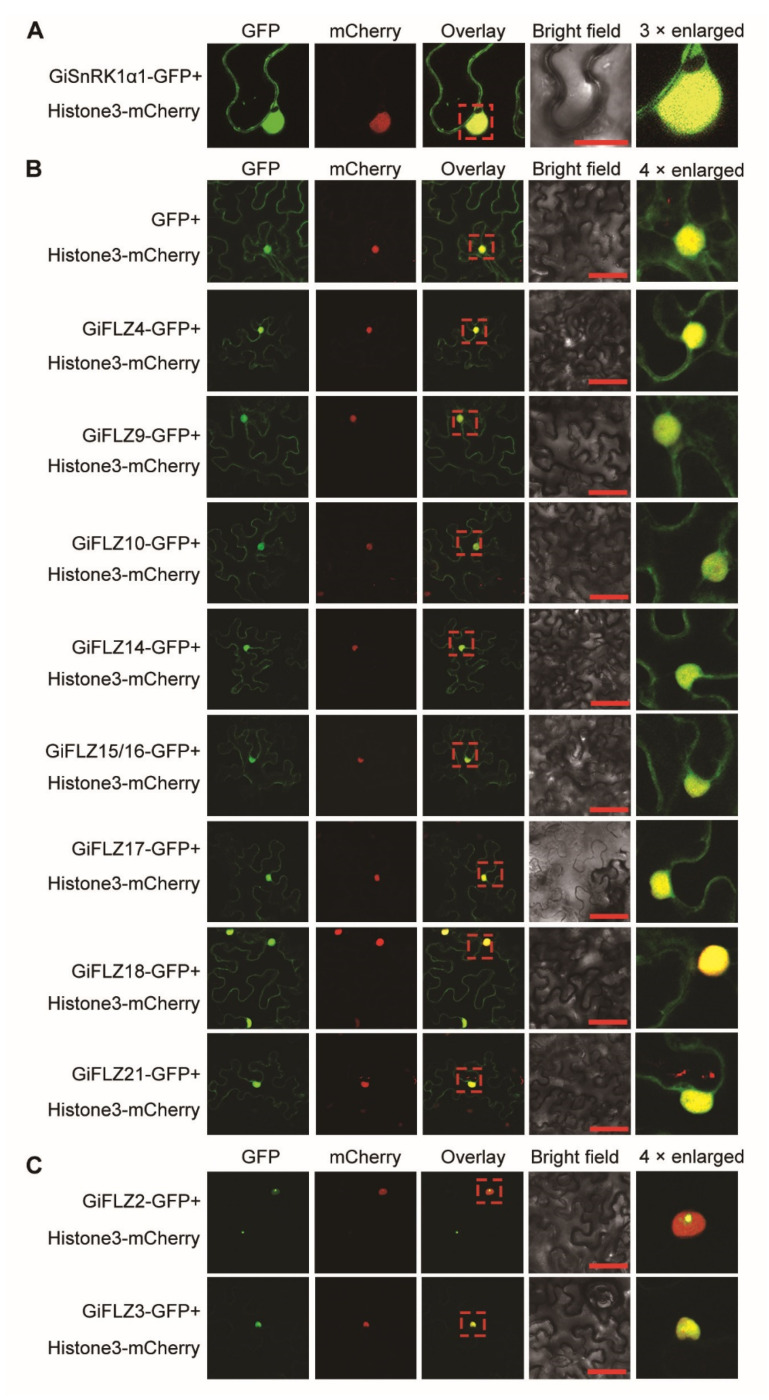
Subcellular localization of GiSnRK1α1 and GiFLZs. GiSnRK1α1-GFP (**A**) and GiFLZ-GFP fusion constructs or empty GFP (**B**, **C**) were transiently expressed in tobacco epidermal cells, respectively. Histone3-mCherry was used as a nuclear marker. Fluorescent images of GFP and mCherry were captured with a confocal scanning microscope and shown in green and red, respectively. Scale bars = 25 μm in (**A**), and 50 μm in (**B**,**C**). The red boxes indicate the enlarged corresponding regions.

**Figure 4 ijms-24-00121-f004:**
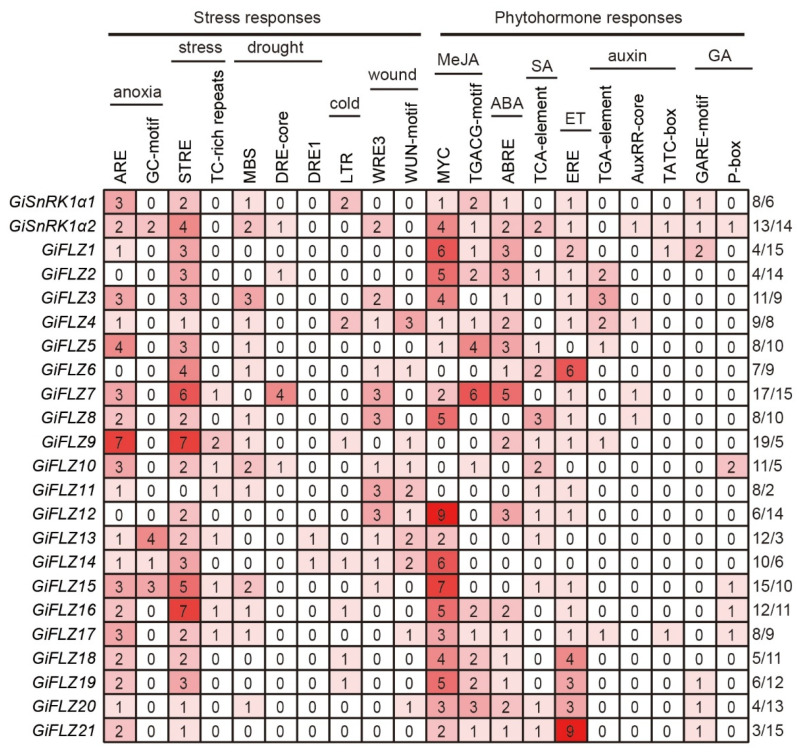
Cis-regulatory element analysis of *GiSnRK1α* and *GiFLZ* gene promoters. The gradient colors in the red grid indicate the number of cis-acting elements in putative promoter regions of the *GiSnRK1α* and *GiFLZ* genes. A white background behind the numbers means there were no cis-elements found in this category. The deeper the red color, the higher number of cis-elements. Numbers at the end of each line indicate the numbers of cis-element found for stress responses (left, before slash) and phytohormone responses (right, after slash), respectively.

**Figure 5 ijms-24-00121-f005:**
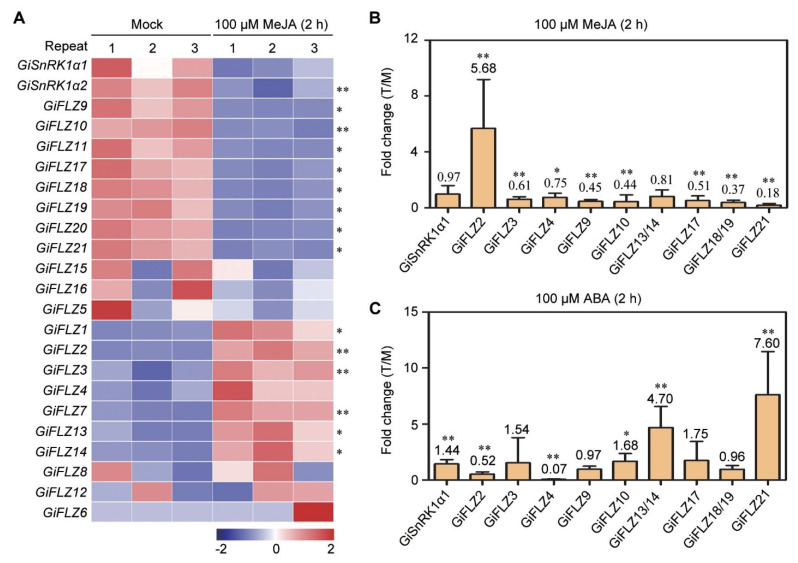
Expression pattern of *GiSnRK1α* and *GiFLZ* genes in response to MeJA and ABA treatment. (**A**) Heat map of the gene expression of *GiSnRK1α* and *GiFLZ* in hairy roots upon 100 μM MeJA treatment. These expression data were normalized by taking log2 of FPKM and are shown from low (blue) to high (red) for each gene. (**B**,**C**) qRT-PCR assay showing the expression of selected *GiSnRK1α* and *GiFLZ* genes in roots of 5-day-old seedlings exposed to 100 μM MeJA (**B**) or 100 μM ABA (**C**) treatment for 2 h. The fold change of treatment (T) to mock (M) are presented as mean ± SD (*n* = 9), and the data on the graph indicate the ratios. *, *p* < 0.05; **, *p* < 0.01—determined by Student’s *t-*test.

**Figure 6 ijms-24-00121-f006:**
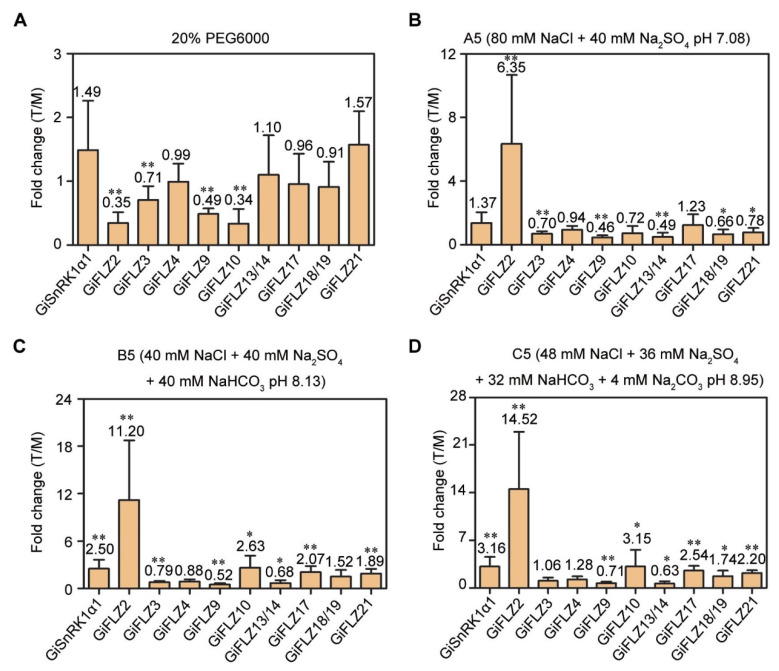
Expression pattern of selected *GiSnRK1α* and *GiFLZ* genes in abiotic stress treatments. The qRT-PCR data were assayed for the expression of selected *GiSnRK1α* and *GiFLZ* genes in the roots of 5-day-old seedlings subjected to 20% PEG6000 (**A**), A5 (**B**), B5 (**C**), and C5 (**D**) treatment for 2 h. The fold change of treatment (T) in relation to mock (M) are presented as mean ± SD (*n* = 9), and the data on the graph indicate the ratios. * *p* < 0.05; ** *p* < 0.01—determined by Student’s *t-*test.

**Figure 7 ijms-24-00121-f007:**
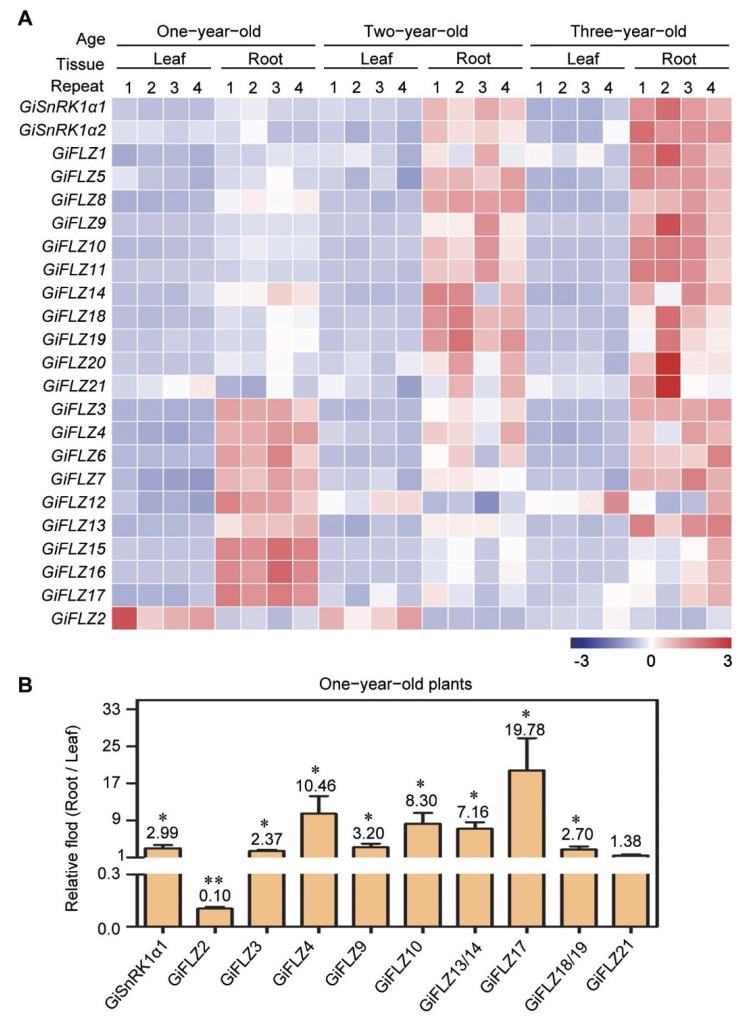
Expression pattern of *GiSnRK1α* and *GiFLZ* genes in roots and leaves. (**A**) Heat map of the gene expression of *GiSnRK1α* and *GiFLZ* in roots and leaves from different *G. inflata* plants of different ages. Data were normalized by taking a 2-based log of FPKM, shown from low (blue) to high (red) for each gene. (**B**) Tissue-specific expression of selected *GiSnRK1α* and *GiFLZ* genes by qRT-PCR assay. The relative expression from root to leaf is presented as mean ± SD (*n* = 3), and the data on the graph indicate the ratios. * *p* < 0.05; ** *p* < 0.01—determined by Student’s *t-*test.

**Table 1 ijms-24-00121-t001:** Summary of GiSnRK1α and GiFLZ members.

Gene Name	Locus ID	Chr ^a^	ORF ^b^ Length (bp)	Protein Length (aa)	MW ^c^ (kDa)	Isoelectric Point	Subcellular Localization ^d^
*GiSnRK1*α*1*	evm.model.8.608	8	1545	514	58.76	8.51	cyto, nucl
*GiSnRK1*α*2*	evm.model.2.1049	2	1545	514	58.71	8.21	cyto, nucl
*GiFLZ1*	evm.model.1.209	1	348	115	13.00	8.92	nucl, chlo
*GiFLZ2*	evm.model.1.568	1	573	190	21.92	10.48	nucl, cyto_nucl
*GiFLZ3*	evm.model.1.4145	1	1338	445	48.91	6.30	nucl, cyto_nucl
*GiFLZ4*	evm.model.1.6486	1	489	162	18.56	8.21	nucl, chlo
*GiFLZ5*	evm.model.1.6930	1	879	292	32.66	5.92	nucl, mito
*GiFLZ6*	evm.model.1.7429	1	381	126	13.93	9.12	chlo, nucl
*GiFLZ7*	evm.model.1.7634	1	711	236	25.90	8.31	nucl, chlo
*GiFLZ8*	evm.model.1.9018	1	471	156	17.96	7.40	nucl, mito
*GiFLZ9*	evm.model.2.1302	2	927	308	33.57	6.87	nucl, cyto
*GiFLZ10*	evm.model.2.2351	2	858	285	31.29	7.70	nucl, chlo
*GiFLZ11*	evm.model.3.1265	3	891	296	32.42	7.91	mito, chlo
*GiFLZ12*	evm.model.3.1977	3	1134	377	41.22	4.82	cyto,nucl
*GiFLZ13*	evm.model.4.583	4	768	255	28.20	8.75	nucl, chlo
*GiFLZ14*	evm.model.4.604	4	762	253	27.99	8.75	nucl, cyto
*GiFLZ15*	evm.model.4.3359	4	450	149	16.80	9.03	chlo, nucl
*GiFLZ16*	evm.model.4.3497	4	450	149	16.80	9.03	chlo, nucl
*GiFLZ17*	evm.model.4.3645	4	1227	408	44.73	6.25	cyto, nucl
*GiFLZ18*	evm.model.5.2797	5	486	161	17.72	10.23	nucl, mito
*GiFLZ19*	evm.model.5.2819	5	495	164	17.94	10.23	nucl, chlo
*GiFLZ20*	evm.model.6.206	6	513	170	18.93	9.89	nucl, chlo
*GiFLZ21*	evm.model.6.799	6	588	195	21.91	8.50	nucl, mito

^a^: chromosome, ^b^: open reading frame, ^c^: molecular weight, ^d^: predicted by WoLF PSORT web server.

## Data Availability

The genomic data that support the findings of this study are available from the National Genomics Data Center (https://ngdc.cncb.ac.cn/; NGDC; CRA009044; accessed on 25 November 2022). The transcriptome data are available from the NCBI GEO repository (http://www.ncbi.nlm.nih.gov/geo; PRJNA574093; accessed on 25 September 2019).

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
