# Peer review of "Genome-Wide Identification of SnRK1 Catalytic α Subunit and FLZ Proteins in Glycyrrhiza inflata Bat. Highlights Their Potential Roles in Licorice Growth and Abiotic Stress Responses"

_ijms, 2022, doi:10.3390/ijms24010121_

Round 1
Reviewer 1 Report
I read the manuscript entitled “Genome-wide identification of SnRK1 and FLZ genes in licorice highlights their roles in plant growth and abiotic stress responses". In this study, the authors analyzed gene expression analysis, Polygenetic analysis, protein interaction, sub-cellular localization, and some other bioinformatics analysis. But some important analyses are missing, such as gene structure, Chromosomal Distribution, Motif Analyses, and cis-regulatory elements. The manuscript needs major revision.
1. In the title, please write the full name of the licorice and check, whether it will italics or not.
2. please check the whole manuscript the gene will be italic and the protein name will be nonitalic, such as line 35, etc.
3. keywords, change the abiotic stress word, write other words
4. Introduction and whole manuscript change the reference format according to the journal format [1]
5. Line 95, don't need to write results in the introduction, In this study, two SnRK1 genes and 21 FLZ genes were identified in G. inflata, a widely
6. Please clearly write your objective end of the introduction.
7. Please improve the whole material and methods part, read some recently published papers, and improves your M&M.
Hussain, Q.; Zheng, M.; Chang, W.; Ashraf, M.F.; Khan, R.; Asim, M.; Riaz, M.W.; Alwahibi, M.S.; Elshikh, M.S.; Zhang, R.; et al. Genome-Wide Identification and Expression Analysis of SnRK2 Gene Family in Dormant Vegetative Buds of Liriodendron chinense in Response to Abscisic Acid, Chilling, and Photoperiod. Genes 2022, 13, 1305. https://doi.org/10.3390/ genes13081305
8. The author used different online websites, software’s etc. But haven’t written the accessed date, and references. E.g, tair , bio-edit software, etc.
9. Table 1 Summary of GiSnRK1 and GiFLZ members, mentioned 13 genes , don’t understand why 13 genes. And what the table 1 showed?
10. line 136, for statistical analysis and Subcellular localization, Yeast two-hybrid assay need references.
11. Result 3.1, according there are two SnRK1 genes in Arabidopsis but total is three. How the author said two genes and when the author accessed the online website.
12. Line 161, the author cited table 1, and 2 SnRK1 and 21 FLZ genes, total 23 but in table total genes are 13.
13. Expression analysis results and figures are totally confused. Please clearly explain and present.
14. Discussion parts no need to repeat the introduction and your results. Please improve the discussion part.
Author Response
Response: Thank you for your time in reviewing our manuscript and your constructive suggestions. We have added the analyses of gene structure, chromosomal distribution, and cis-regulatory elements in the revised manuscript. We re-organized the whole manuscript and asked native speaker in plant bio field to edit the language. Please find the point-to-point response below:
- In the title, please write the full name of the licorice and check, whether it will italics or not.
Response: We have corrected it accordingly.
- please check the whole manuscript the gene will be italic and the protein name will be nonitalic, such as line 35, etc.
Response: We have double-checked themanuscript and corrected these errors.
- keywords, change the abiotic stress word, write other words
Response: We have revised the keywords.
- Introduction and whole manuscript change the reference format according to the journal format [1]
Response: We have changed the reference format.
- Line 95, don't need to write results in the introduction, In this study, two SnRK1 genes and 21 FLZ genes were identified in G. inflata, a widely
Response: We have deleted these sentences.
- Please clearly write your objective end of the introduction.
Response: We have added the information.
- Please improve the whole material and methods part, read some recently published papers, and improves your M&M.
Hussain, Q.; Zheng, M.; Chang, W.; Ashraf, M.F.; Khan, R.; Asim, M.; Riaz, M.W.; Alwahibi, M.S.; Elshikh, M.S.; Zhang, R.; et al. Genome-Wide Identification and Expression Analysis of SnRK2 Gene Family in Dormant Vegetative Buds of Liriodendron chinense in Response to Abscisic Acid, Chilling, and Photoperiod. Genes 2022, 13, 1305. https://doi.org/10.3390/ genes13081305
Response: Thank for your kind reminder. We have revised this part according to your suggestion.
- The author used different online websites, software’s etc. But haven’t written the accessed date, and references. E.g, tair , bio-edit software, etc.
Response: We have added the related information.
- Table 1 Summary of GiSnRK1 and GiFLZ members, mentioned 13 genes, don’t understand why 13 genes. And what the table 1 showed?
Response: We are sorry for this. We submitted a word file and format was changed when the system generated the PDF file. We uploaded a PDF file for the revised version and hopefully the fill will show up correctly.
- line 136, for statistical analysis and Subcellular localization, Yeast two-hybrid assay need references.
Response: We have added the requested information.
- Result 3.1, according there are two SnRK1 genes in Arabidopsis but total is three. How the author said two genes and when the author accessed the online website.
Response: We have revised the description in the introduction section in detail.
- Line 161, the author cited table 1, and 2 SnRK1 and 21 FLZ genes, total 23 but in table total genes are 13.
Response: We are sorry for this. We submitted a word file and format was changed when the system generated the PDF file. We uploaded a PDF file for the revised version and hopefully the fill will show up correctly.
- Expression analysis results and figures are totally confused. Please clearly explain and present.
Response: We have reorganized the results section and added detail descriptions in the figure legends.
- Discussion parts no need to repeat the introduction and your results. Please improve the discussion part.
Response: We have reorganized discussion part.

Reviewer 2 Report
This manuscript includes a lot of experiments and has a lot of data. However, I didn't see the criteria to confirm the SnRK1s and FLZs in G. inflata and the classification of FLZs, and the writing of the article is also rough. Many parts are not clearly described, inconsistent, and there are some problems in the experiments. I suggest that the author rearrange the data, remove the inaccurate experiments, and re-write and re-submit it with a better writing idea. It is better to ask native English experts to check the language before submission.
line 78-81, According to your work, and SnRK1 in plants actually encodes a catalytic subunit of a heterotrimeric protein, you have to make a explanation to avoid the misunderstanding. And we can't find the reference of "Chen et al., 2017" in references.
line 103, you have to provide the genome data of G. inflata before the submission
line 105, roots and leaves "and" under MeJA?
line 123, salinity 160 mM what?
line 124, one sample should be one biological replicate.
line 202-203, Are you sure? Cytoplasmic membrane or cystoplasm?
Author Response
Reviewer 2
This manuscript includes a lot of experiments and has a lot of data. However, I didn't see the criteria to confirm the SnRK1s and FLZs in G. inflata and the classification of FLZs, and the writing of the article is also rough. Many parts are not clearly described, inconsistent, and there are some problems in the experiments. I suggest that the author rearrange the data, remove the inaccurate experiments, and re-write and re-submit it with a better writing idea. It is better to ask native English experts to check the language before submission.
Response: Thank you for your time in reviewing our manuscript and your constructive suggestions. We have reorganized the results and a native English speaker (who is also an expert in plant biology field) has edited the language thoroughly.
line 78-81, According to your work, and SnRK1 in plants actually encodes a catalytic subunit of a heterotrimeric protein, you have to make a explanation to avoid the misunderstanding. And we can't find the reference of "Chen et al., 2017" in references.
Response: Thank you for kind suggestion. We have added the detail information of SnRK1 in the introduction section and checked the reference carefully.
line 103, you have to provide the genome data of G. inflata before the submission
Response: We have uploaded the genome data to National Genomics Data Center (https://ngdc.cncb.ac.cn/;NGDC; CRA009044).
line 105, roots and leaves "and" under MeJA?
Response: We have corrected this mistake.
line 123, salinity 160 mM what?
Response: We have revised this part.
line 124, one sample should be one biological replicate.
Response: Yes, we harvested five roots and mixed them as one sample or one biological replicate. Each experiment was done with three biological replicates.
line 202-203, Are you sure? Cytoplasmic membrane or cystoplasm?
Response: We carefully analyzed the confocal images and concluded that SnRK1-GFP is localized to the nucleus and cytoplasm. As we know, the central vacuole is large in the mature tobacco leaves cells, so the cytoplasm space is very condensed and looks like cytoplasmic membrane. And this subcellular localization pattern is mimic to that of AtSnRK1.1/1/2 (Regulation of Sucrose non-Fermenting Related Kinase 1 genes in Arabidopsis thaliana. Front Plant Sci. 2015, 5, 324).

Reviewer 3 Report
Article with a relevant theme and relevant scientific interest.

Round 2
Reviewer 1 Report
The author addressed all my comments and suggestion. And the author also has done suggested analyses such as gene structure, Chromosomal Distribution, Motif Analyses, and cis-regulatory elements. Now the author just needs to write about gene structure, Chromosomal Distribution, Motif Analyses, and cis-regulatory elements in material and methods. Please carefully check all the figures and tables number in the main text, including supplementary files. I suggested accepting the article after minor revisions.
Author Response
Thanks for your time and kind suggestions. We have added more detailed informaiton about gene structure analysis. Minor revisions were also made. Please see more details in the manuscript file.